# HIF-1α-Mediated Disruption of Cellular Junctions: The Impact of Hypoxia on the Tumor Microenvironment and Invasion

**DOI:** 10.3390/ijms26115101

**Published:** 2025-05-26

**Authors:** Michael Springer, Zeynep Aydin Burakgazi, Anastasiia Domukhovska, Ben Nafchi, Michael C. Beary, Arielle Acquisto, Juliette Acquisto, Vladyslav Komarov, Madison Jensen, Brandon Gulledge, Maksym Poplavskyi, Md Gias Uddin, Gamal Rayan, Shoshanna N. Zucker

**Affiliations:** D’Youville University School of Pharmacy, Buffalo, NY 14201, USA; sprinm22@dyc.edu (M.S.); aydinz08@dyc.edu (Z.A.B.); domuka28@dyc.edu (A.D.); alimob19@dyc.edu (B.N.); bearym23@dyc.edu (M.C.B.); acquia29@dyc.edu (A.A.); acquij29@dyc.edu (J.A.); komarv03@dyc.edu (V.K.); jensem29@dyc.edu (M.J.); gulleb18@dyc.edu (B.G.); poplam01@dyc.edu (M.P.); uddinm@dyc.edu (M.G.U.); rayang@dyc.edu (G.R.)

**Keywords:** hypoxia, gap junctions, adherens junctions, tight junctions, desmosomes, matrix metalloproteinases, tumor microenvironment, angiogenesis, metastasis

## Abstract

Hypoxia is a critical factor affecting tissue homeostasis that dramatically alters the tumor microenvironment (TME) through genetic, metabolic, and structural changes, promoting tumor survival and proliferation. Hypoxia-inducible factor-1α (HIF-1α) plays a central role in this process by regulating hundreds of genes involved in the processes of tumorigenesis, angiogenesis, metabolic reprogramming, and immune evasion. This review provides a comprehensive examination of the role of HIF-1α in hypoxia and how hypoxia weakens intercellular junctions—including gap junctions, adherens junctions, tight junctions, and desmosomes. The disruption of gap junctions decreases intercellular communication, which alters signal transduction cascades and tumor suppressive properties. Adherens junctions are comprised of proteins that characterize the tissues and link cells to the actin cytoskeleton, whereby their disruption promotes the epithelial-to-mesenchymal transition (EMT). Under hypoxic conditions, the tight junction proteins are dysregulated, altering paracellular transport and cell polarity. In addition, desmosomes provide linkage to intermediate filaments, and hypoxia compromises tissue integrity. Collectively, the influence of hypoxia on cellular junctions promotes tumorigenesis through reducing cell communication, cytoskeletal interactions, and altering signaling pathways. Activation of matrix metalloproteinases (MMPs) further degrades the extracellular matrix and enhances tumor invasion and metastasis. This process also involves hypoxia-induced angiogenesis, regulated by HIF-1α. A comprehensive understanding of the mechanisms of hypoxia-driven tumor adaptation is essential for developing effective therapeutic strategies. Furthermore, this review examines current treatments aimed at targeting HIF-1α and explores future directions to enhance treatment efficacy and improve patient outcomes.

## 1. Introduction

Hypoxia in tumor cells is a key feature in many solid and aggressive cancers, which arises from an inadequate oxygen consumption-to-oxygen delivery ratio in the tissue [1,2]. As the tumor cells divide, their metabolic demands increase, significantly outpacing the oxygen supply provided by existing vasculature. Although neovasculature aims to compensate for the oxygen and nutrient shortages, these new blood vessels are often dysfunctional and inefficient, leaving the oxygen delivery to remain inadequate [3,4]. This oxygen shortage creates a hypoxic tumor microenvironment (TME), which acts as a major cellular stress factor. To adjust to the hypoxic TME, cells activate a series of adaptive reactions that augment the cell survival and proliferation [1,2,3]. As a result, a hypoxic TME not only plays a critical role in tumor progression but also profoundly alters cellular interactions with the surrounding cytoskeleton, vasculature, and immune cells by disrupting cell–cell communication, particularly through the downregulation of intercellular junctions [2].

One of the most critical consequences of hypoxia is its ability to disrupt intercellular junctions. Gap junctions are essential for maintaining intercellular communication, while tight junctions regulate paracellular transport. Adherens junctions and desmosomes maintain tissue structure and integrity by anchoring cells to one another and to the cytoskeleton. Hypoxia-induced changes, particularly those mediated by HIF-1α, destabilize these structures by downregulating, modifying, or degrading junctional proteins. In cancer, this disruption promotes epithelial-to-mesenchymal transition (EMT), cytoskeletal remodeling, increased permeability, and ultimately, metastasis [5].

This review provides a comprehensive analysis of how hypoxia influences cancer cell behavior by examining hypoxia-induced adaptations within the tumor microenvironment (TME), with particular emphasis on the effects on cellular junctions (Figure 1). It focuses on the extensively studied HIF-1α subunit, a key mediator of the hypoxic response, and its downstream effects on intercellular communication. Additionally, the review explores molecular processes that were recognized to be influenced by alterations in gap junction intercellular communication (GJIC), such as angiogenesis, metabolic reprogramming, cell migration and invasion, and immune evasion—key processes that drive tumor progression within the hypoxic TME.

## 2. The Molecular Basis of Hypoxic Signaling

### 2.1. HIF-1α: Master Regulator of Hypoxic Response in Cancer

HIF1-α is the oxygen-sensitive subunit of the HIF-1 transcription factor that orchestrates cellular adaptation to hypoxia. It plays a critical role in cancer progression and therapy resistance by regulating gene expression changes that drive cancer progression, through processes such as angiogenesis, metabolism, cell survival, and metastasis [2,4]. By mediating hypoxia-adaptive pathways, HIF1-α enables cancer cells to successfully survive in low-oxygen conditions, making it a major target for cancer therapeutics development.

The protein structure of HIF-1α includes a basic helix–loop–helix domain near the C-terminal, two distinct PAS (PER-ARNT-SIM) domains, a PAC (PAS-associated C-terminal) domain, a nuclear localization signal motif, two transactivating domains (CTAD and NTAD), and an intervening inhibitory domain. The PAS domains, which facilitate protein–protein interactions, enable HIF-1α to dimerize with HIF-1β (also known as ARNT) and form a heterodimeric complex, which then binds to hypoxia-response elements (HREs) in the promoter regions of target genes to regulate hypoxic adaptation [6,7,8,9].

The regulation of HIF-1α is tightly controlled at the post-translational level through several modifications, including hydroxylation, ubiquitination, and proteasomal degradation. Under normoxic conditions, HIF-1α is rapidly degraded via the ubiquitin–proteasome pathway. In the presence of oxygen, prolyl hydroxylase domain (PHD) enzymes catalyze the hydroxylation of specific proline residues on HIF-1α. The hydroxylated HIF-1α is then recognized by the von Hippel–Lindau (VHL) tumor suppressor protein, which targets it for ubiquitination and subsequent proteasomal degradation [8,9].

However, under hypoxic conditions, the activity of PHD enzymes is inhibited due to the lack of oxygen, enabling HIF-1α to evade hydroxylation, which leads to the stabilization and accumulation of HIF-1α levels. Once stabilized, HIF-1α translocates to the nucleus, where it dimerizes with HIF-1β and binds to hypoxia-response elements (HREs) in the promoter regions of target genes to activate their transcription [10,11,12,13,14,15]. After binding, the HIF-1 complex recruits transcriptional coactivators, such as CBP/p300, which help it sustain the gene expression and enhance transcriptional activity [16,17].

HIF-1α controls the expression of over 300 genes involved in key cellular processes, including the regulation of the genes that drive tumor growth. By upregulating vascular endothelial growth factor (VEGF) and other pro-angiogenic factors, it promotes the growth of neovasculature to supply tumors with the nutrients needed for their growth and metastasis [7,8]. To further improve oxygen delivery to hypoxic areas, HIF-1α also induces erythropoietin production, which stimulates the production of red blood cells, thereby ensuring a sustained oxygen supply to the tumor region.

### 2.2. Cross-Talk with PI3K-Akt, MAPK, and ROS

Hypoxia drives the formation of new blood vessels by activating molecular pathways that stimulate angiogenesis while simultaneously suppressing the immune response. A central player in this process is a pro-angiogenic factor, Vascular Endothelial Growth Factor (VEGF), along with its receptors (VEGFRs), which are strong mediators of neovasculature formation and are upregulated in most cancers [18,19]. In addition to VEGF, hypoxia also induces the expression of inducible Nitric Oxide Synthase (iNOS), leading to increased production of Nitric Oxide (NO). NO is a critical mediator of hypoxia-induced angiogenesis, working with VEGF to promote neovascularization. In addition, VEGF modulates the immune response [20,21].

HIF-1 tightly regulates VEGF, such that under low-oxygen conditions, its subunit HIF-1α stabilizes and binds to the VEGF gene, triggering its expression [22]. VEGF primarily interacts with VEGFR-2, a receptor on endothelial cells, to promote their development, migration, and increased vascular permeability [23]. Through VEGFR-2 activation, VEGF stimulates key signaling pathways, such as PI3K-Akt and MAPK, which enhance cell survival and proliferation. By sustaining blood vessel formation, tumors ensure a continuous supply of oxygen and nutrients, supporting the growth of cancer cells [24]. The neovasculature in tumors is often abnormal and leaky, yet it still facilitates oxygen and essential nutrient delivery to the tumor site. As cancer cells rapidly divide, their metabolic demands increase, exacerbating hypoxia and further stimulating angiogenesis in an attempt to sustain the oxygen and nutrient supply. This self-perpetuating cycle results in a tissue that remains highly hypoxic despite its extensive yet inefficient vascular network [19,24].

## 3. Hypoxia-Induced Disruption of Cell–Cell Junctions

Beyond aiding oxygen delivery, HIF-1α drives cancer cell invasion and metastasis. It achieves that by promoting epithelial-to-mesenchymal transition (EMT) and influencing the expression of genes that remodel the extracellular matrix and support cell migration [2,6,9,25,26]. Among the structures that are most affected by hypoxia are cellular junctions, which include gap junctions, adherens junctions, tight junctions, and desmosomes [8]. These intercellular junctions play a key role in maintaining communication between cells, cell–cell adhesion, and tissue integrity. They can be easily compromised, mislocalized, and degraded under hypoxic conditions due to HIFs activating genes that destabilize intercellular channels and suppress connexin expression as part of an adaptation response to hypoxia. By weakening intercellular channels, HIFs promote cellular detachment, tumor invasion, and metastasis [9,27]. Notably, these changes have been shown to be reversible, making hypoxia-induced disruption of cellular junctions a promising therapeutic target for restoring homeostasis within the TME [5,28]. A model of the changes between cellular junctions in normoxia and hypoxia is outlined in Figure 2.

### 3.1. Gap Junctions: Structure and Function

Gap junctions function as intercellular communication channels, which play a critical role in maintaining coordinated tissue function. They allow the transfer of molecules up to 1 kDa, including amino acids, ions, nutrients, metabolites, and signaling molecules. By maintaining direct cell–cell signal exchange, they enable cells to respond collectively to the changes in the microenvironment, including the spread of stress signals and apoptotic cascades [29,30]. Importantly, gap junction permeability is not static, and it dynamically responds to the changes in the cellular environment by adjusting the composition of its connexins appropriately [30,31].

Gap junctions act as channels with permeability that vary based on their connexin composition [30]. Connexins are the protein subunits that form connexons, with six connexins forming a connexon. Two connexons from adjacent cells dock together to form the gap junction [31]. These hexameric assemblies form a 2-nanometer extracellular gap, ensuring high selectivity and allowing only small ions and molecules to pass through.

Connexins are composed of intracellular N- and C-termini, extracellular E1 and E2 loops, and four transmembrane domains [32]. The E1 and E2 loops connect transmembrane domains (T1–T2 and T3–T4), while a cytoplasmic loop spans the intracellular space between T2 and T3. Connexins are classified by predicted mass into five subfamilies—α, β, γ, δ, and ε in mice and GJA, GJB, GJC, GJD, and GJE in humans. Each subfamily has distinct biophysical properties and expression patterns, contributing to functional diversity [31].

There are 21 members of the connexin (Cx) family in humans, which are designated either by their molecular weight (i.e., Cx43) or their nomenclature based on sequence similarity and the ability to heterodimerize (i.e., Cx43 is GJA1, which forms heterodimers with other alpha connexins, whereas beta connexins such as Cx26 and Cx32 can heterodimerize) [33,34,35]. There is a great deal of tissue specificity between connexin family members, as well as diversity in signaling capabilities [33,34,36].

The gap junction specificity is partially dictated by the phosphorylation sites in the C-terminal domain. Cx43 has sites in the C-terminal region, which are phosphorylated by MAPK and v-src [37,38], which regulates gap junction closure or gating properties. In contrast, Cx26 has a very short C-terminal domain and thus is not regulated by phosphorylation-based gating. Cx43 gap junctions play an important role in tumor progression, in which they are highly characterized as tumor suppressors. However, in advanced states of tumor metastasis, Cx43 can actually promote tumor invasion. This is largely due to interactions within the C-terminal domain, as well as hemichannel functions [39].

Oxidative stress can mediate the selectivity of transport in gap junctions. Bagati, et al. demonstrated that increased exogenous expression of Cx43 in melanoma enhanced the apoptotic effects of non-thermal plasma, likely through an upregulation of Cx26 and the bystander effect of reactive oxygen and nitrogen species (RONS) transport [40]. It has recently been demonstrated that the permeability of the Cx26 hemichannels for water and RONS molecules is higher in the presence of oxidized lipids [41]. Thus, the selectivity of transport through gap junctions and channel gating properties are important regulatory mechanisms for intercellular communication, which can be affected by tumor hypoxia.

#### 3.1.1. Hypoxia and Gap Junctions

The variety of permeants that can pass through GJs during gap junction intercellular communication (GJIC) and hemichannel transport dictates whether the channels promote viability or cell death, usually associated with apoptosis. Permeants such as ATP and cAMP promote cell viability, whereas pro-apoptotic signals, such as IP3 and Ca^++^, promote cell death [42,43]. GJIC can either promote cell growth or inhibit it, depending on the cellular context and the stage in tumorigenesis [41]. Hypoxia can hinder the spread of pro-apoptotic signaling between damaged and neighboring cells, which, in cancer, manifests as uncontrolled growth and aberrant survival of the cells isolated from intercellular communication. This isolation from neighboring influence allows subpopulations of cancer cells to adapt independently to the hypoxic TME, gain selective advantages, and contribute to intratumoral heterogeneity [44,45].

Heterogeneity creates subpopulations with varying invasion potential. Under hypoxia, these differences result in heterogeneous EMT states, with a spectrum of epithelial and mesenchymal phenotypes of high plasticity. Such plasticity allows tumor cells to switch between states in response to environmental conditions. As a result, compromised junctional integrity enhances both adaptability and metastatic potential, allowing highly motile tumor cells to easily breach the basement membrane and invade surrounding tissues [46,47].

In intratumoral heterogeneity, oxygenation disparities also lead to varying interactions with signaling molecules such as growth factors. As a result, some subpopulations undergo proliferation, while others remain quiescent. This specific effect of tumor heterogeneity ultimately impairs treatment efficacy, making it more challenging to target the tumor cells due to distinct adaptive mechanisms of cell subpopulations to therapies [18,44,45,48,49].

Under hypoxic conditions, cells frequently modulate GJIC by altering connexin expression, a response that affects metabolic substrate sharing among neighboring cells. Early evidence from astrocytes demonstrated that reduced oxygen tension upregulates Cx43 expression and alters GJIC, influencing lactate and glucose exchange to support cell viability [50]. Later work highlighted similar phenomena in other cells, including cardiomyocytes, where preserving gap junctions during chronic hypoxia helped maintain ATP levels [50]. In breast cancer cells, downregulation of Cx43 during hypoxia was tied to heightened glycolytic flux, underscoring a strong link between limited oxygen supply, dampened GJIC, and metabolic reprogramming [31].

More recently, Kutova et al. have discussed how connexins play multifaceted roles in the tumor microenvironment, affecting not just direct cell-to-cell communication but also oxygen gradients and metabolic adaptations that sustain tumor growth [32]. Indeed, broader analyses confirm that an altered connexin profile (and, by extension, altered gap junction functionality) can be a critical component in tumor initiation and maintenance under hypoxia [51].

#### 3.1.2. Mechanisms of Hypoxia-Induced Gap Junction Alterations

##### Transcriptional Regulation of Connexin Genes

Hypoxia leads to a reduction in connexin expression, particularly connexin 43 (Cx43), through HIF-1α-mediated transcriptional repression. When HIF-1α binds to hypoxia-responsive elements (HREs) in the promoter regions of target genes, transcriptional regulation is altered. In the case of connexin genes, this results in decreased mRNA synthesis and lowered protein levels. Reduced connexin expression weakens the formation and functionality of gap junctions, impairing intercellular communication [47,52].

##### Post-Translational Modifications

Under hypoxic conditions, increased production of reactive oxygen species (ROS) leads to oxidative stress, which directly affects connexin proteins. Key post-translational modifications that pertain to gap junction destruction during hypoxia include phosphorylation, which alters the gating properties of gap junction channels, reducing their ability to remain open. In addition, the processes of nitrosylation and oxidation destabilize the connexin structure, making it more susceptible to degradation. These modifications disrupt the assembly and functionality of gap junction channels, further impairing GJIC [34].

##### Gap Junction Channel Dysfunction

Connexins are highly sensitive to changes in the calcium concentration, as calcium is involved in the activation of protein kinases, which then phosphorylate connexins. Excessive calcium levels and subsequent phosphorylation lead to conformational changes that close the channels. Thus, hypoxia-induced elevation of intracellular calcium levels contributes to the closure of gap junction channels, which reduces the ability of cells to exchange critical signaling molecules [53].

Notably, hypoxia also interferes with the intracellular trafficking of connexins to the plasma membrane, such that instead of being transported to their functional locations, connexins may accumulate in the Golgi apparatus or instead become targeted for lysosomal degradation. This reduction in functional gap junctions at the cell surface leads to diminished intercellular signaling and coordination, further compromising proper tissue functioning [32,54].

### 3.2. Adherens Junctions

In addition to gap junctional regulation, adherens junctions play a major role in tissue integrity, and their disruption during hypoxia can lead to changes in the epithelial-to-mesenchymal transition (EMT). The primary function of adherens junctions is to provide strong attachments between epithelial cells, ensuring proper tissue integrity. These junctions consist of protein complexes anchored to the actin cytoskeleton, and the key proteins that make up adherens junctions are cadherins and catenins (β, α, and *γ*) [55].

Epithelial cadherin (E-cadherin) establishes intercellular contact through homophilic interactions with other E-cadherin proteins on neighboring cells. Additionally, E-cadherin interacts with catenins, which play essential roles in nucleocytoplasmic trafficking, transcriptional regulation, and ubiquitination. This interaction enables anchoring of the adherens junction to the actin cytoskeleton in the cytoplasm [56,57,58].

Other important proteins in adherens junctions include vinculin, nectin, and afadin. Vinculin maintains the structure and stability of the adherens junction by binding to α-catenin and β-catenin, acting as a mechanical stabilizer and linking cell adhesion complexes to the actin network. Nectin, in complex with afadin, also connects the adherens junction to the cytoskeleton through the facilitation of cell–cell adhesion. By trans-dimerizing with nectins on the neighboring cells, as well as recruiting E-cadherin to contact sites, nectin further strengthens cell adhesion [59,60].

E-cadherin also undergoes trans-dimerization with neighboring cells, leading to the fusion of clusters and the formation of a mature adherens junction. Once established, the adherens junction binds to actin through α-catenin, which crosslinks the actin filaments, stabilizing the junction within the cytoskeleton. Overall, this complex structure enables adherens junctions to maintain strong cell–cell connections. The importance of these proteins was demonstrated through afadin knockout experiments, which ultimately led to the disorganization of cell junctions. Additionally, inhibiting nectin dimerization has been shown to prevent cadherin-dependent adherens junction formation [56].

#### 3.2.1. Hypoxia-Induced Adherens Junction Alterations

Under hypoxic conditions, exosomes in prostate cancer cells express a higher number of proteins than in normoxia, which are primarily associated with remodeling of the adherens junction pathways. This increases invasiveness and stemness, inducing TME changes, thereby promoting tumor aggressiveness [61]. Interestingly, in some cases, HIF-2α can regulate adherens junctions and cell polarity, inducing the TME even in conditions of normoxia [62].

In parallel with changes to gap junctions, low oxygen frequently disrupts adherens junction integrity, primarily by downregulating E-cadherin. Studies in ovarian carcinoma cells, for instance, have shown that hypoxia promotes an E-cadherin-to-N-cadherin switch, correlating with a metabolic transition toward glycolysis and increased invasive capacity [58]. Similarly, in other cancer models, hypoxic activation of transcription factors such as ZEB1 or β-catenin not only drives EMT and the breakdown of E-cadherin-based adhesions but also shifts cells toward a more glycolysis-dominant energy strategy [63,64]. These observations echo findings from a 2020 review on hypoxia-induced EMT, which emphasizes that low oxygen availability accelerates the loss of epithelial characteristics, including adherens junction integrity, while concomitantly upregulating glycolytic enzymes [65].

Even beyond cancer, broader regulatory mechanisms have shown that hypoxia can synergize with other pathways to downregulate adherens junction proteins, reinforcing the concept that diminished cell–cell adhesion under low oxygen is intimately coupled to metabolic reprogramming [26].

#### 3.2.2. Adherens Junctions and EMT

Adherens junctions play a vital role in epithelial–mesenchymal transition (EMT), a process in which epithelial cells lose their apical–basal polarity and weaken their cell–cell junctions. EMT drives tissue morphogenesis during embryonic development and facilitates wound healing by closing open wounds. However, when dysregulated, EMT can be exploited by cancer cells to promote metastasis. Several transcription factors promote EMT, including Snail (SNAI1), TWIST1, and ZEB. Snail represses the expression of E-cadherin, thereby leading to the disassembly of adherens junctions. This is a critical step in EMT process, as it allows cells to detach from the epithelium and migrate to other areas [60,66,67].

Adherens junctions regulate EMT through two main pathways: endocytosis and component recycling, which occur both via clathrin-mediated and non-clathrin-mediated mechanisms. Dysregulation of these pathways in cancer disrupts the balance between recycling and degradation, leading to rapid loss of E-cadherin and increased cell migration. An important oncogene involved in this process is Src, which phosphorylates tyrosine residues on E-cadherin, displacing p120-catenin and activating signaling pathways that inhibit E-cadherin expression, ultimately promoting tumor progression [66,67,68].

Overall, adherens junctions play a crucial role in maintaining cell-to-cell adhesion by forming a strong scaffold that holds cells together and anchors them to the cytoskeleton. The proteins that make up adherens junctions, such as E-cadherin, undergo modifications during EMT to facilitate essential processes like wound healing under normal conditions. However, when dysregulated, EMT can contribute to cancer progression [57,59,60,66]. Understanding the mechanisms of EMT and adherens junctions, as well as their interplay, can provide insights into potential cancer treatment strategies.

### 3.3. Tight Junctions

Changes in cell adhesion and polarity also affect epithelial cell transformation and metastatic capacity. In addition to adherens junctions, tight junctions play key roles in cell-to-cell adhesion, polarity, and migration. Tight junctions are composed of various proteins such as claudins, which help maintain cell adhesion and regulate signal transduction, cell growth, and migration through EMT [69]. Their regulation is influenced by multiple factors, including angiogenesis, glycolysis, and hypoxic conditions. Among these, hypoxia is a primary contributor to defects in tight junction proteins.

Under low-oxygen conditions, the upregulation of HIF-1 disrupts multiple signaling pathways involved in the regulation of tight junction proteins [69,70,71]. Specifically, hypoxia can increase JAM3 expression, decrease Par3 levels, and alter the transcription of various claudin proteins [70,72]. These changes compromise tight junction integrity, ultimately promoting cancer metastasis.

Hypoxia-induced signaling disruptions can downregulate claudin proteins; in cancer, claudins—particularly claudins 3, 4, and 7—tend to be dysregulated. Changes in their expression, both downregulation and upregulation, can weaken cell adhesion and alter the structure and function of tight junctions [71,73]. When tight junctions and their associated proteins lose proper function, cell motility increases, a process that cancer cells exploit during metastasis [74].

CLDN1 is a key claudin in tight junctions that regulates multiple signaling pathways involved in cell invasion and migration. Due to its broad regulatory function, CLDN1 is influenced by various pathways, including regulation by ADAM15 through the PI3K/Akt/mTOR pathway, which is associated with breast cancer signaling [75,76]. Additionally, upregulation of CLDN1 has also been observed to activate the ERK signaling pathway in the MCF7 breast cancer cell line, further emphasizing its role in cancer progression. These findings highlight the important role of claudins in cellular signaling pathways and demonstrate how disruptions in these pathways can contribute to tumor formation and metastasis [76,77].

Tight junctions also contain Par3 protein, which plays a crucial role in aligning the apical barrier of cells. Reduced levels of Par3 have been associated with increased metastasis in cancer cells [69,71]. Similarly, JAM3, another tight junction protein, contributes to the regulation of EMT. Overexpression of JAM3 has been observed in cancers characterized with excessive EMT activity, underlining the impact of tight junction components on tumor advancement [72,74]. In cervical cancer, JAM3 overexpression activated the HIF-1α/VEGFA pathway, promoting cell migration and invasion, whereas silencing of JAM3 inhibited these processes [72]. Collectively, these results indicate potential drug targets by overexpression of Par3 or downregulation of JAM3 [72].

#### Hypoxia-Induced Tight Junction Alterations

A hypoxic lung cancer-secreted exosomal miRNA, miR-23a has the potential to upregulate HIF-1α in endothelial cells, thereby increasing angiogenesis, as well as vascular permeability, in cancer cells through the inhibition of tight junction protein, ZO-1 [78]. CLDN4 overexpression is associated with HIF-1α and VEGFA, demonstrating a worse prognosis in patients with advanced non-small cell lung cancer [79]. Another tight junction protein involved in metastatic progression is MARVELD3, which is downregulated in Snail-induced EMT in the progression of pancreatic cancer [80]. The hypoxia-induced increase in HIF-1α in lung cancer cells also causes a decrease in Claudin-5 expression, which increases the permeability of the blood–brain barrier, promoting lung cancer metastases to the brain [81].

### 3.4. Desmosomes

Desmosomes are another important structure that promotes cellular adhesion, linking cells through intermediate filaments. They function as adhesive junctions that couple cells through desmosomal cadherins, which include desmogleins (DSGs) and desmocollins. These cadherins mediate calcium-dependent cell-to-cell adhesion, which is essential for desmosomal function [82,83].

The human genome encodes four DSGs, which are single-pass transmembrane proteins that bind to the intermediate filaments, desmoplakin and plakoglobin, within the cell cytoplasm and confer the attachment between cells [84]. Among desmosomal genes, DSG2 is particularly affected by hypoxia and is dynamically regulated by the oxygen levels in the TME, playing a key role in promoting metastatic processes [77,85]. In hypoxic tumor regions, DSG2 is downregulated, which triggers EMT-related gene expression changes in primary tumors. EMT enables cancer cells to detach, undergo intravasation, and circulate throughout the body. As tumor cells exit hypoxic conditions and undergo reoxygenation, DSG2 expression is reinstated, enabling them to successfully colonize distant organs [77,86].

DSG2 has five extracellular tandem cadherin domains, including the second extracellular domain (EC2), to which a peptide (KC21) has been shown to inhibit angiogenesis and retinal neovascularization [87]. This result suggests a potential target for angiogenesis in tumor progression.

#### Hypoxia-Induced Desmosome Alterations

HIF-1α serves as a key regulator of DSG2 repression in hypoxia by recruiting EZH2 and SUZ12, components of the Polycomb Repressive Complex 2 (PRC2). Their binding to the DSG2 promoter results in sustained DSG2 repression, a process correlated with poor prognosis and increased recurrence risk, particularly in breast cancer patients [77,86,88]. In addition to desmogleins, desmoplakin (DSP) is also affected by hypoxia. The recent discovery of a compound, N^6^-methyladenosine (m^6^A), showed it is demethylated in more aggressive pituitary neuroendocrine tumors. This demethylation destabilizes DSP mRNA, which affects desmosome integrity and enhances tumor hypoxia tolerance [89], suggesting a therapeutic targeting mechanism.

### 3.5. Alterations in Junctional Proteins and Cell Polarity

Intercellular junctional proteins are essential to regulatory mechanisms within tissues that maintain homeostasis, and thus the disruption of the tissue integrity during hypoxia promotes tumor formation. A model of cellular changes induced by hypoxia in tumorigenesis is outlined in Figure 3. Normal epithelial cells maintain cell polarity with an epithelial surface and a basolateral side. The proteins that comprise cell junctions contribute to this cellular polarity, and thus their altered regulation in hypoxia affects processes related to tissue organization; linkage to the cytoskeleton; and in the cases of gap junctions, intercellular transport, and in tight junctions, paracellular transport. Studies have shown that disruption of cellular polarity can promote tumorigenesis [14,34,90]. Gap junctions have a particularly complex pattern of regulation, leading to their dual functionality roles as both tumor suppressors and promoters of metastasis. Due to the ability of gap junctions in combination with adherens junctions, tight junctions, and desmosomes to promote tissue integrity, it is apparent as to why their disruption and altered regulation promotes cancer development. However, the complexity of the intercellular signaling combined with the intracellular signaling pathways provides a further analysis of the mechanisms of cellular transformation.

Gap junction regulation during carcinogenesis is very complex and tissue specific. While gap junctions are often characterized as tumor suppressors, they are often upregulated in tumor metastasis [33]. Due to this altered pattern of regulation, it is often difficult to target gap junctions with inhibitors to affect tumorigenesis. In addition to GJIC and transport of ions and metabolites into the extracellular milieu through hemichannels [91], there are also non-GJIC functions of gap junctions that affect the process of tumorigenesis.

The C-terminal domain of Cx43 has been shown to bind to microtubules and regulate microtubule dynamics, which help to maintain cell stability [92,93]. In addition, hemidesmosomes bind to intermediate filaments such as keratin in the basement membrane. The disruption of cell polarity by the loss of cell junctional attachments has been shown to cause a disassembly of the hemidesmosomes. Following hemidesmosome disassembly, α6 β4 integrin becomes phosphorylated and relocates to F-actin-rich protrusions, where integrins interact with actin filaments, which promotes cell invasion [94,95]. This is also associated with the degradation of the ECM and promotion of EMT.

## 4. Pathophysiological Consequences of Junctional Breakdown

### 4.1. MMPs and ECM Remodeling

As cellular junctions disassociate during hypoxia, individual cells undergo EMT and form new junctions with mesenchymal tissue, promoting cellular metastasis. Overexpression of matrix metalloproteinases (MMPs) in the TME disrupts ECM homeostasis, leading to enhanced communication between neighboring tumor cells and promoting malignant processes such as cancer invasion, migration, and metastasis [14,96]. Hypoxia not only influences MMP activity directly by upregulating or downregulating their function but also modulates their activity indirectly by affecting the expression of tissue inhibitors of metalloproteinases (TIMPs) [97]. A hypoxic environment has been associated with decreased levels of TIMPs in the TME, leading to an imbalance in the MMP-to-TIMP ratio. This disruption enhances MMP activity at the tumor site, making ECM degradation more likely and promoting metastatic activity within the tissue [97,98]. MMPs have also been shown to act on non-matrix substrates, such as chemokines, growth factors and receptors, adhesion molecules, and apoptotic mediators [97,99]. These interactions elicit cellular responses that provide an environment optimal for tumor progression and, in combination with the ability of MMPs to degrade ECM proteins, contribute to the creation of a microenvironment that heavily supports cancer cell metastasis [98].

Hypoxia within the TME drives ECM remodeling through the activation of HIFs, significantly altering the structure and function of ECM components [70]. Acting as a physical scaffold, the ECM ensures proper tissue structure, and additionally supports key processes like migration, differentiation, proliferation, and apoptosis by interacting with integrins and modulating the cell behavior [1,6]. Under normal physiological conditions, the ECM undergoes controlled remodeling as part of essential processes, such as development and tissue repair, with proteolytic activity terminated once the necessary process is complete [2,8]. However, in pathological conditions such as hypoxia, this balance is disrupted due to activation of HIFs [70]. These factors, particularly HIF-1α, not only directly enhance the transcription of MMPs but also indirectly contribute to MMP upregulation by stimulating neighboring host cells to secrete cytokines and growth factors, resulting in further overproduction of MMPs [100]. The overexpression of MMPs leads to excessive proteolysis in the ECM, where these enzymes degrade and reorganize ECM components, causing localized alterations in the alignment of ECM fibers [2,14]. These structural changes disturb tissue homeostasis, fostering a microenvironment favorable to tumor progression [13,97].

### 4.2. Invasion and Migration

Invasion and migration of cancer cells, which are the key steps towards establishing metastasis, rely heavily on ECM remodeling, basement membrane degradation, as well as EMT [25,98]. EMT is facilitated within the TME due to the stabilization of HIF-1α levels by hypoxic conditions, which drives the expression of EMT-associated transcription factors, such as Snail, Twist, and ZEB [98,101]. These factors suppress epithelial markers like E-cadherin while upregulating mesenchymal markers like N-cadherin and vimentin, which leads to a significant reduction of cell-to-cell adhesion, giving the cells an ability to detach from the primary tumor site, invade surrounding tissue, and initiate metastatic dissemination [59,99]. The process of how hypoxia induces angiogenesis to promote metastasis is depicted in Figure 4.

Consequently, once they invade, mesenchymal–epithelial transition (MET) enables the cancer cells to regain epithelial traits, anchor, and proliferate at a new, secondary site. This intricate interplay between the two transitions allows tumor cells to successfully metastasize and establish secondary tumors, driving cancer progression and complicating treatment strategies. The ability of tumor cells to proliferate at secondary sites arises from the process of reoxygenation, which occurs when they leave the hypoxic environment of the primary tumor site. Reoxygenation disrupts the previously stable levels of HIF-1α, reversing EMT and enabling MET. By reacquiring an epithelial phenotype, these cells regain the ability to anchor and multiply, ultimately forming secondary tumors [60,97]. However, rapid proliferation at the secondary tumor site reestablishes the hypoxic environment and stimulates angiogenesis.

### 4.3. The Role of iNOS and NO in Tumor Progression

In hypoxic conditions, iNOS expression is upregulated, leading to increased production of nitric oxide (NO), a molecule with a complex and dual role in cancer progression. The impact of NO depends on its concentration, its duration of exposure to cells, as well as TME conditions [102]. At controlled levels, NO promotes angiogenesis by increasing vascular permeability and amplifying VEGF activity. However, excessive NO synthesis can be cytotoxic, highlighting the importance of tight regulation of iNOS activity [103,104]. In most cancers, iNOS upregulation has been associated with pro-tumorigenic effects, where it promotes metastasis through TME remodeling [102,103,104,105]. However, under certain conditions, such as excessive NO concentrations that induce oxidative stress or immune activation, iNOS overexpression has been linked with better clinical outcomes [105]. Despite these context-dependent anti-tumorigenic effects, iNOS overexpression is more commonly associated with pro-tumorigenic effects, including angiogenesis, immune suppression, and metastasis [104].

### 4.4. COX-2 as a Driver of Angiogenesis and Inflammation

COX-2, an inflammatory enzyme, is frequently overexpressed in tumors. In hypoxic conditions, it promotes angiogenesis by enhancing the effects of VEGF [106]. Furthermore, by inhibiting anti-cancer immune responses and enhancing the activity of regulatory T cells, COX-2 aids further tumor development by helping it evade the immune system. Its overexpression has been associated with more aggressive tumors and poorer prognoses, particularly in colorectal cancer. Selective COX-2 inhibitors, such as celecoxib, have shown promise in reducing tumor growth and angiogenesis, especially when combined with VEGF-targeting therapies [107]. COX-2, iNOS, and VEGF operate in a feedback loop to sustain angiogenesis and drive tumor progression. VEGF induces the synthesis of iNOS, which then raises NO levels and subsequently promotes angiogenesis and modulates the immune microenvironment to support cancer progression [107]. Simultaneously, iNOS interacts with COX-2 to create a pro-inflammatory, tumor-promoting environment. COX-2 not only amplifies VEGF signaling but also increases the production of pro-angiogenic prostaglandins, further reinforcing continuous angiogenesis [107,108]. These intricate interactions between various pathways underscore the necessity of combination therapies that target multiple pathways to effectively disrupt this cycle [109].

### 4.5. Immunological Impact of Hypoxia in the Tumor Microenvironment

TME hypoxia significantly influences immune responses by suppressing anti-tumor immunity and altering immune cell function, thereby promoting tumor growth and fostering treatment resistance [110,111]. HIFs regulate the production of various immune-related molecules and induce metabolic reprogramming in tumor cells, generating acidic byproducts that inhibit immune cell activity and reduce the capacity for immune surveillance [85].

Noman and coauthors reported that HIF-1α directly drives the expression of programmed cell death ligand (PD-L1), thus helping tumor cells evade immune surveillance. They further showed that blocking PD-L1 under hypoxic conditions can re-energize T-cell activity [112]. HIFs further decrease anti-tumor immune responses by recruiting and activating immunosuppressive cells, such as regulatory T cells and myeloid-derived suppressor cells [70,85,113].

The impact of hypoxia extends beyond tumor cells, directly compromising the activity of immune effectors. Low oxygen levels impair the function of cytotoxic T lymphocytes (CTLs) and natural killer (NK) cells, reducing their cytolytic activity, survival, and proliferation [113]. Additionally, hypoxia disrupts dendritic cell maturation and antigen presentation, hindering the activation of adaptive immune responses [112].

These adaptations protect tumor cells from immune-induced cell death and create an immunosuppressive TME that not only supports tumor growth and metastasis but also greatly complicates the efficacy of immunotherapies. Thus, targeting hypoxia-induced pathways presents a promising strategy to improve therapeutic outcomes. Preclinical and early clinical studies have demonstrated the potential of combining hypoxia-modulating therapies with immune checkpoint inhibitors to overcome hypoxia-induced immunosuppression. Such synergistic approaches may help restore immune cell function and improve patient outcomes [114]. Understanding the mechanisms by which hypoxia and HIFs drive immune evasion is essential for the development of innovative strategies to counteract these pathways and enhance the efficacy of immunotherapies [85,110].

## 5. Therapeutic Implications and Approaches

### 5.1. Resistance to Chemotherapy

Hypoxia facilitates tumor resistance and diminishes the efficacy of chemotherapy through several mechanisms. Hypoxic tumors often possess disorganized and leaky vasculature that limits the delivery of chemotherapeutic agents to the tumor core. This uneven drug distribution enables hypoxic cells to evade cytotoxic effects, which then often results in treatment failure [1,6,7]. Hypoxia also upregulates the expression of drug efflux pumps, such as P-glycoprotein, via pathways mediated by HIF-1α, which actively transport drugs out of cells and reduce intracellular drug concentrations, thereby diminishing treatment efficacy [24,115].

Moreover, hypoxic tumor cells undergo metabolic reprogramming by shifting to anaerobic glycolysis and producing lactate as a byproduct. This metabolic adaptation reduces reliance on mitochondrial respiration, which renders therapies targeting oxidative metabolism less effective. This adaptation also supports the survival of cancer stem-like cells, which are inherently resistant to chemotherapy [24,116].

Thus, by altering the TME, hypoxia creates an immunosuppressive stroma, which shields tumor cells from immune-mediated cytotoxicity. This alteration reduces the efficacy of immunomodulatory chemotherapy and further complicates treatment strategies [4,24].

### 5.2. Resistance to Radiation Therapy

While radiation therapy relies on the generation of RONS to induce DNA damage in cancer cells, hypoxia reduces the oxygen tension, thus impairing this process. RONS formation is diminished under hypoxic conditions and results in the reduced efficacy of radiation therapy. In fact, tumor cells in hypoxic regions have been shown to be up to three times more resistant to radiation compared to normoxic cells [110,115,117].

Beyond limiting RONS generation, hypoxia also induces signaling pathways that enhance the expression of DNA repair proteins and enable tumor cells to efficiently repair radiation-induced DNA damage. HIF-1α mediates this process by activating genes involved in homologous recombination and non-homologous end-joining pathways [115,116,117]. Additionally, hypoxia induces cell cycle arrest by halting cells in the G1 phase, which is less susceptible to radiation-induced cytotoxicity, through the upregulation of cyclin-dependent kinase inhibitors, such as p21 and p27 [4,24,117].

Since tumor hypoxia fluctuates spatially as well as temporally, it creates pockets of radioresistant cells. This heterogeneity complicates treatment planning while significantly reducing the efficacy of standard radiation therapy [4,118]. Radiation has been associated with the upregulation of gap junctions and, more specifically, with GJIC. This enables the transport of RONS through gap junctions to promote enhanced cell death [119]. Similarly, non-thermal plasma induces Cx43 gap junctions to enhance transport of RONS by the bystander effect [40]. Clinically, this presents a potential treatment for both promoting RONS delivery and enhancing gap junctions to promote tissue integrity.

### 5.3. HIF-1α as a Therapeutic Target

HIF-1α is an attractive therapeutic target since this subunit stands as a master regulator of the cellular hypoxic response, orchestrating a complex adaptive mechanism in cancer biology. Several approaches can be explored, which include, but are not limited to, direct HIF-1α inhibitors, which are small molecules that may target HIF-1α stability and/or transcriptional activity; upstream pathway inhibitors, which are compounds that modulate PHDs or other regulatory proteins; downstream effector targeting, in which case molecules inhibit HIF-1α target genes and their products; or combination approaches, where HIF-1α-targeted therapies are integrated, leading to the improved efficacy of conventional treatments to overcome hypoxia-induced resistance [70,99,120,121].

### 5.4. Therapy Aimed at Gap Junction Restoration

Efforts to mitigate the effects of hypoxia on gap junctions include the use of connexin mimetic peptides, which are synthetic peptides that mimic the function of connexins and are able to restore GJIC under hypoxic conditions. Another approach involves targeting HIF pathways to prevent the transcriptional downregulation of connexins [52]. Additionally, antioxidants also help preserve gap junction function by preventing connexin degradation, protecting cells from hypoxia-induced loss of GJIC. Emerging research further highlights the role of non-coding RNAs, such as microRNAs, in regulating connexin expression under hypoxia, presenting new promising therapeutic targets.

### 5.5. Therapy Directed at Reversing the EMT

The EMT has a high degree of plasticity, enabling a reversion of this process [5]. Epigenetic modifications, including DNA methylation, histone modification, chromatin remodeling, and non-coding RNAs, regulate the state of EMT and tumor progression [28]. Many metastasis-related miRNAs target genes that have established roles in metastasis, including MMPs and E-cadherin [122]. Because epigenetic changes can be reversed, this also lends itself to new mechanistic approaches for drug targeting [28]. Hypoxia has been shown to induce EMT in prostate cancer through upregulating EMT-Activator Zeb1 and SK3 Channel Expression [123]. As more mechanisms of hypoxia-induced alterations in the EMT are identified, the number of targets will exponentially increase.

### 5.6. Current Therapeutic Strategies

There are several treatment strategies that have emerged within target hypoxia-driven tumor biology, especially the HIF-1α pathway and its downstream mediators. These approaches are at varying stages of clinical development and may offer promising insights for restoring junctional integrity, mitigating EMT progression, as well as improving responsiveness to therapeutic intervention.

Anti-angiogenic agents such as bevacizumab have been successfully utilized in standard treatment regimens for several solid tumors, which include colorectal, lung, and renal cell carcinomas. Bevacizumab enhances oxygen delivery by normalizing the tumor vasculature and reducing HIF-1α stabilization, which indirectly results in preserved epithelial junctions and the suppression of EMT-mediated invasion [124,125].

Belzutifan is a HIF-2α inhibitor that was originally FDA approved for von Hippel–Lindau (VHL)-associated renal cell carcinoma and now represents a more direct therapeutic strategy. It is undergoing further studies in clear-cell RCC, as well as other hypoxia-driven malignant disease states. Belzutifan’s ability to inhibit HIF-2α transcriptional activity enables it to downregulate the expression of genes that promote angiogenesis, metabolic reprogramming, and loss of junctional proteins. Clinical data from the LITESPARK trials show improved tolerability and disease control, making HIF-2α inhibition a viable component of biomarker-driven treatment regimens [126].

Immune checkpoint inhibitors, such as nivolumab and pembrolizumab, have shown enhanced efficacy when they are combined with agents that relieve tumor hypoxia, such as anti-angiogenics and HIF inhibitors. This synergistic effect is being explored in early-phase trials, which demonstrate the reversal of hypoxia-induced T-cell exclusion and the restoration of immune surveillance [127].

Epigenetic modulators, such as the histone deacetylase inhibitors (HDACs) vorinostat and entinostat, are currently under clinical evaluation for their ability to reverse EMT phenotypes. These agents can re-induce the expression of junctional proteins inhibited by hypoxic stress like E-cadherin and claudins, and have been trialed in breast, pancreatic, as well as non-small cell lung cancers. HDAC inhibitors that have been paired with hypoxia-targeted therapies may potentiate structural re-differentiation of tumor tissue [128].

The clinical deployment of these agents is increasingly guided by molecular profiling, such as in hypoxia gene signatures, VHL mutation status, and EMT scoring indices. The integration of these biomarkers into treatment algorithms enables more effective therapeutic selection, as well as the ability for combinations of strategies that address the multifactorial consequences of hypoxia-induced junctional disruption.

## 6. Conclusions and Future Directions

Hypoxia and HIF-1α signaling are key components of tumor progression, as well as resistance to therapeutic interventions; the translation of these insights calls for more integration of hypoxia-targeted strategies into current clinical practice. The FDA approval of belzutifan represents a key advancement, which validates the therapeutic inhibition of the HIF pathways, which provide opportunities for similar agents with even broader applications. Future efforts should focus on expanding the indications for HIF inhibitors beyond VHL-associated tumors to define their role in sporadic cancers.

Strategies that combine hypoxia modulation with immune checkpoint inhibition, epigenetic reprogramming, or anti-angiogenic therapy may hold tremendous potential in overcoming microenvironment-driven therapeutic resistance. These approaches must be personalized using integrated molecular diagnostics to maximize the clinical benefit, using hypoxia imaging, transcriptomic profiling, as well as EMT biomarker panels to accomplish this.

Restoring junctional integrity and reversing the molecular sequelae of hypoxia represents a novel, as well as actionable, therapeutic axis. The continued refinement of biomarker-driven treatment models will be pivotal in guiding the optimal deployment of these therapies in clinical oncology.

An understanding of the molecular basis of how hypoxia disrupts the cellular junctions that leads to EMT, enhanced invasion, angiogenesis, and metastasis will allow us to develop new therapeutic strategies, particularly those involved in the regulation of proteins that comprise gap junctions, adherens junctions, tight junctions, and desmosomes. Targeting these mechanisms that decrease hypoxia-driven cellular communication and integrity mechanisms not only holds the potential to inhibit tumor progression but also to overcome hypoxia-induced therapeutic resistance.

Research that is focused on clarifying the intricate relationships between junction proteins and their downstream signaling pathways will help to elucidate novel targets for drug design. Identifying synergistic approaches that affect the immune system as well as EMT progression has the potential to enhance treatment efficacy. Additionally, advances in biomarker-based therapeutic strategies, particularly those reflecting hypoxia-driven processes, may further enable more individualized and effective cancer treatments.

## Figures and Tables

**Figure 1 ijms-26-05101-f001:**
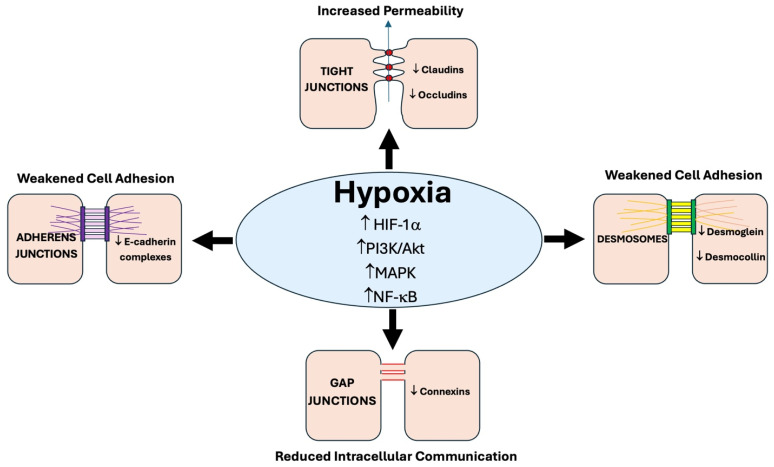
Schematic overview of the effects of hypoxia on various cellular junctions. Central hypoxia conditions activate the Hypoxia Inducible Factor (HIF-1α) pathway, the PI3K/Akt pathway, the Mitogen-Activated Protein Kinase (MAPK) pathway, and the Nuclear Factor kappa-light-chain-enhancer of activated B cells (NF-κB) pathway. These pathways lead to alterations in tight junctions, adherens junctions, gap junctions, and desmosomes. The activation pathways are indicated by the upward arrows, whereas downregulation pathways are indicated by downward arrows. Collectively, these changes contribute to increased cellular permeability, weakened cell adhesion, and altered cell–cell communication.

**Figure 2 ijms-26-05101-f002:**
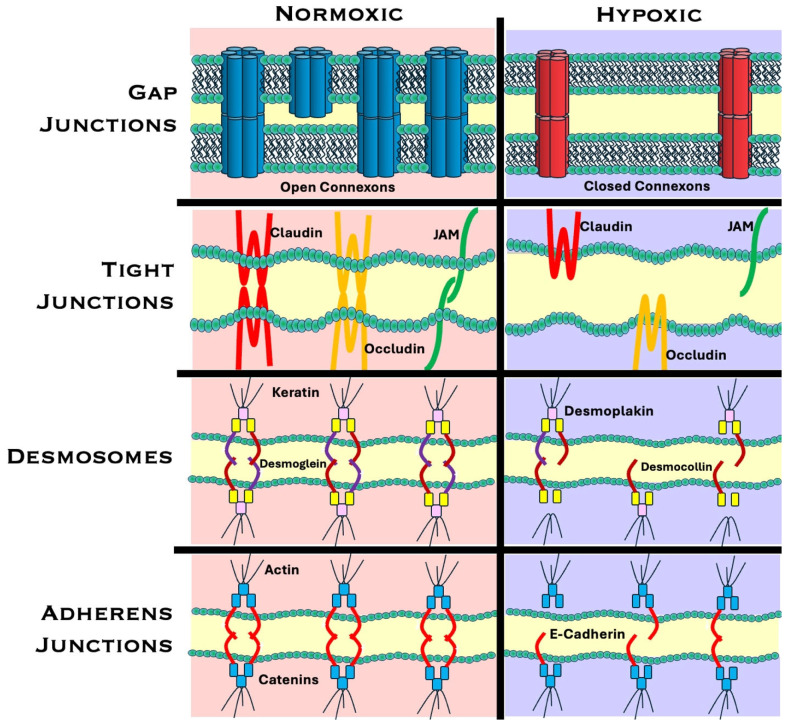
The disruption of cellular junctions under hypoxic conditions is caused by changes to the structure, function, and expression of junctional proteins. In gap junctions, GJA1 and GJA5 are suppressed, reducing the expression of Cx43 and Cx40 connexons; ROS-driven changes affect the remaining connexons, which are internalized or degraded; high cytosolic Ca^2+^ levels lead to closing of connexon hemichannels; the intercellular space between cells increases, and the communication between hypoxic cells decreases, promoting motility. In tight junctions, HIF-1α alters claudin and JAM expression, increasing CLDN 1, 3, and 4 and JAM3 and decreasing CLDN 5 and 7; increased iNOS activity phosphorylates occludins, which are internalized, and since hypoxia stalls Rab13-/Rab11-dependent recycling, they accumulate in the Golgi instead of returning to the membrane; the combined effect of these changes causes leaky membranes, promoting invasion and metastasis. In desmosomes, DSG2 and DSG3/DSC2 (variably) are transcriptionally repressed, which decreases expression of desmoglein; desmoplakin (DSP) is affected by the demethylation of N^6^-methyladenosine (m^6^A), which destabilizes DSP mRNA; these changes promote invasion and metastasis. In adherens junctions, CDH1 (E-cadherin) is silenced, and N-cadherin and VE-cadherin rise; detachment from E-cadherin by MMP-3/7 or ADAM10/15 frees β-catenin; sustained loss of E-cadherin propagates EMT in a wave across the tumor rim.

**Figure 3 ijms-26-05101-f003:**
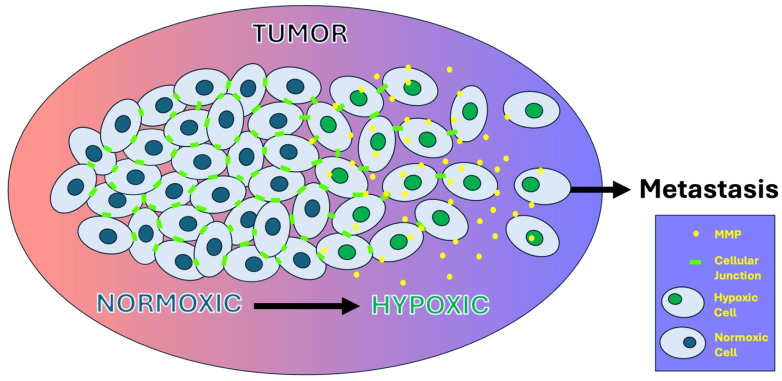
The tumor microenvironment is often marked by increasing hypoxic conditions due to the rapid proliferation of cancer cells outpacing existing vasculature. Increasing hypoxia triggers a change in the nucleus, leading to (among other things) increased synthesis of matrix metalloproteinases (MMPs), which disrupt cellular junctions. Disruption of cellular junctions by MMPs in hypoxic cancer cells increases the risk of metastasis.

**Figure 4 ijms-26-05101-f004:**
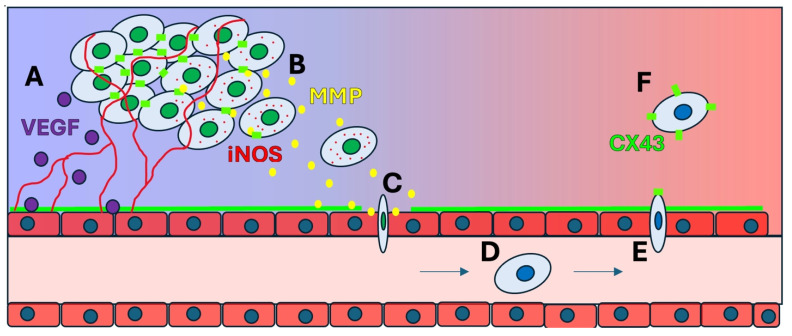
How the disruption of cellular junctions affects invasion and metastasis. (**A**) HIF-1α stabilization drives transcription of VEGF in tumor and stromal cells, promoting angiogenesis and vascularization; (**B**) HIF-1α promotes increased production of iNOS, which sustains hypoxic conditions, and MMP, which disrupts cellular junctions, including CX43 connexin, and promoting invasion; (**C**) Extracellular MMP destabilizes the ECM and basement membrane, promoting intravasation; (**D**) Increased COX-2 and iNOS increase PGE_2_ and NO levels, which recruit platelets to “cloak” the tumor cell in transit in vasculature; (**E**) The hypoxia-induced increase in VEGF, iNOS, COX-2, and PGE_2_ from the tumor cell weakens junctions in the vasculature and promotes extravasation at a distant microvessel; (**F**) Reoxygenation occurs in the new tissue, HIF-1α wanes, and E-cadherin and CX43 reappear, increasing cellular adhesion and promoting metastasis.

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
