# Peer review of "HIF-1α-Mediated Disruption of Cellular Junctions: The Impact of Hypoxia on the Tumor Microenvironment and Invasion"

_ijms, 2025, doi:10.3390/ijms26115101_

Round 1
Reviewer 1 Report
Comments and Suggestions for Authors
The review, by Domukhovska et al. in its present form, does not have a scientific focus and presents too many topics in a quite superficial ways.
The Authors must revise the structure of the review paying attention to keep a narrow focus on the topic the want critically review and paying attention to the the hierachic organization of paragraphs and subsections.
The title suggests that the Authors want to review the effects of HIF-1alpha on cell junction disruption. However, then, they reserve a very little part of the manuscript to this interesting topic, that I suggest to expand. This part, at the moment, is not supported by an adequate number of citations. For the rest, the manuscript alternates didascalic description of the cellular junction, MMP and EMT mechanisms, without offering a significant contribution to the field. Then the manuscript shift to the description of neo-angiogenesis, changing again the focus of the review. Finally, the authors describe the effects of hypoxia on mechanisms related with therapy resistance. Both these parts are not exhaustive, the revision is not systematic and it's not clear the rationale followed by the Authors to select the references cited in the text.
Reviewer 2 Report
Comments and Suggestions for Authors
In this review entitled "HIF-1a-mediated disruption of cellular junctions: The impact of hypoxia on the tumor microenvironment and invasion," the authors discussed the role of hypoxia in the TME and its impact on tumor progression. While the paper provides valuable insights into the role of hypoxia and HIF-1α in tumor progression, the review lacks a clear focus and the structure is somewhat disorganized. The Conclusions and Future Direction section is somewhat vague and lacks depth. In addition, several issues need to be addressed:
- The introduction should briefly mention the key focus areas of the review, such as MMPs and cell junctions.
- Please check the text in Figure 1, as it appears to contain non-English characters.
- The manuscript dedicates an inordinate amount of space to gap junctions and MMPs, while the focus should be on the impact of hypoxia on these structures.
- The descriptions of the mechanisms are overly brief throughout the manuscript. For example, under the section "Mechanisms of Hypoxia-Induced Gap Junction Disruption," each part should be expanded with more detailed explanations to provide a deeper understanding.
- It is recommended to add a summary figure or table for each section to help visually consolidate key points
- In this review is insufficient discussion on the current state of HIF-1α inhibitors or other therapeutic approaches that are in clinical development. additionally, in the Conclusions and Future Direction section, a more detailed discussion on how to better utilize these therapeutic approaches in the clinic should be provided.
Author Response
Thank you for taking the time to review this manuscript. In response to the original version lacking focus and being too disorganized, we have completely rewritten the manuscript to provide a clear focus on the role of hypoxia affecting cellular junctions to promote tumorigenesis and metastasis with a specific focus on the role of HIF-1a and how hypoxia affects each of the junctions (gap junctions, adherens junctions, tight junctions, and desmosomes). I will proceed to provide a response to each of the reviewer's comments:
- The introduction should briefly mention the key focus areas of the review, such as MMPs and cell junctions.
In the introduction, we now clearly state that "
This review provides a comprehensive analysis of how hypoxia influences cancer cell behavior by examining hypoxia-induced adaptations within the tumor microenvironment (TME), with particular emphasis on the effects on cellular junctions (Figure 1). It focuses on the extensively studied HIF-1α subunit, a key mediator of the hypoxic response, and its downstream effects on intercellular communication. Additionally, the review explores molecular processes that were recognized to be influenced by alterations in gap junction intercellular communication (GJIC) such as angiogenesis, metabolic reprogramming, cell migration and invasion, and immune evasion—key processes that drive tumor progression within the hypoxic TME."
- Please check the text in Figure 1, as it appears to contain non-English characters.
The text of Figure 1 has been reviewed and updated to include only English characters.
2. The manuscript dedicates an inordinate amount of space to gap junctions and MMPs, while the focus should be on the impact of hypoxia on these structures.
We have narrowed the focus to address the role of Hypoxia on Cellular junctions. We begin with a detailed description of hypoxia, focusing specifically on the role of HIF-1a. We then proceed to describe the cellular junctions in structure and function for each in sequence, gap junctions, adherens junctions, tight junctions, and desmosomes. Each section is followed by a section on how that junction is altered during hypoxia. The text is accompanied by a new figure which highlights the structure of the junctions in normoxia and how each is altered in hypoxia.
The next section explains how the cellular junctions are altered with the EMT and tumorigenesis. Another section was added which outlines how hypoxia alters the process of tumorigenesis with a focus on changes in cell polarity. This is accompanied by Figure 3 which demonstrates the changes in junctions and MMPs in tumorigenesis.
That section led into the detailed process of how hypoxia influences metastasis with a section on angiogenesis, invasion, and metastasis, including several subsections. These sections are accompanied by Figure 4 which shows a detailed model of the metastatic cascade, focusing on the role of hypoxia and cellular junctions in the Figure legend.
3. The descriptions of the mechanisms are overly brief throughout the manuscript. For example, under the section "Mechanisms of Hypoxia-Induced Gap Junction Disruption," each part should be expanded with more detailed explanations to provide a deeper understanding.
We have added new sections to outline how each of the four junctions is altered in the state of hypoxia, following the description of each of the junctions.
4. It is recommended to add a summary figure or table for each section to help visually consolidate key points
We have expanded the number of figures from two to four outlining the detailed processes of the manuscript.
In Figure 1, we provide an overview of hypoxia affecting the cellular junctions with a focus on HIF-1a.
In Figure 2, we show a model of the structure of the junctions in normoxia and how they are altered in hypoxia.
In Figure 3, we provide a model of how tumorigenesis is affected by the state of hypoxia with a focus on junctions and MMPs.
In Figure 4, we show an outline of the metabolic cascade and how hypoxia induces angiogenesis to promote intravasation, extravasation, and metastasis.
5. In this review is insufficient discussion on the current state of HIF-1α inhibitors or other therapeutic approaches that are in clinical development. additionally, in the Conclusions and Future Direction section, a more detailed discussion on how to better utilize these therapeutic approaches in the clinic should be provided.
We have expanded the final section of the manuscript on therapeutic approaches to include current therapies that are in use and future directions for utilizing these therapies in the clinic.
In summary, we have completely rewritten and reorganized the manuscript since the version that was originally submitted.
Overall, we hope that we have significantly improved this manuscript with a clear focus on the role of hypoxia affecting cellular junctions to promote tumorigenesis and metastasis. We hope to convey to the reviewer that this study now has a significant impact on the field of tumor hypoxia and cellular junctions.
Round 2
Reviewer 1 Report
Comments and Suggestions for Authors
I appreciated the efforts made by the Authors to improve the quality and the readability of the manuscript. The first part of the revised version is now more focused, and, on my honest opinion, more interesting for the readers. The second part, addressing various mechanism of tumorigenesis influenced directly or indirectly by HIF-α and cell junctions disruption remain didascalic and partially unfocused.
Moreover, the organization of the structure of the manuscript is still poor and the organization of the paragraphs and subparagraphs is still confusing. I suggest using consecutive numbers to identify the main chapters of the review organized in 2 or 3 levels of sub-paragraphs, clearly identified by numbering (i.e. 1.1 or 1.1.1 etc.).
A more concise discussion of the second part of the manuscript would improve the overall readability. More critical selection of the literature and identification of clear criteria for the inclusion of the references cited in the text is also expected.
Author Response
I appreciated the efforts made by the Authors to improve the quality and the readability of the manuscript. The first part of the revised version is now more focused, and, on my honest opinion, more interesting for the readers. The second part, addressing various mechanism of tumorigenesis influenced directly or indirectly by HIF-α and cell junctions disruption remain didascalic and partially unfocused.
Thank you for taking the time to review this manuscript. We have made a concerted effort to address the concerns about the manuscript and to revise it accordingly. As per the reviewer's suggestion, we have shortened the second part of the manuscript and rearranged some sections. We have reorganized the order of the manuscript, making a new section under the subheading, "2.2. Cross-talk with PI3K-Akt, MAPK, and ROS" which is at the beginning of the paper following the section "2.1 HIF-1α: Master Regulator of Hypoxic Response in Cancer" in a collective section 2 entitled, "The Molecular Basis of Hypoxic Signaling". We have decreased the section on MMPs and designated an umbrella heading as "4. Pathophysiological Consequences of Junctional Breakdown" which includes "MMPs and ECM Remodeling" among other relevant subheadings to improve the flow of the manuscript. We have removed the section focusing on the effects of hypoxia on different tumor types. We have aligned section 4 with the description of Figure 4, entitled "How the disruption of cellular junctions affects invasion and metastasis". With this reorganization and condensation of the manuscript, we hope to have improved the focus and readability.
Moreover, the organization of the structure of the manuscript is still poor and the organization of the paragraphs and subparagraphs is still confusing. I suggest using consecutive numbers to identify the main chapters of the review organized in 2 or 3 levels of sub-paragraphs, clearly identified by numbering (i.e. 1.1 or 1.1.1 etc.).
We thank the reviewer for this suggestion. We have revised the entire manuscript with numbering for headings and subheadings. We agree that this improves the organization and focus of the manuscript.
A more concise discussion of the second part of the manuscript would improve the overall readability. More critical selection of the literature and identification of clear criteria for the inclusion of the references cited in the text is also expected.
As mentioned above, we have condensed and reorganized the second part of the manuscript at the request of the reviewer. We carefully reviewed the references to see that they are cohesive and aligned with the information in the text. We included pertinent references about the current medications that have been FDA approved to target hypoxia-related mechanisms. Once again, we thank the reviewer for the critical analysis, and we hope that we have satisfied the standards for publication in the International Journal of Molecular Sciences.
Reviewer 2 Report
Comments and Suggestions for Authors
I have carefully examined the authors' responses to my previous comments and the corresponding revisions. The main concerns have been adequately resolved. The current version can be considered for acceptance.
Author Response
I have carefully examined the authors' responses to my previous comments and the corresponding revisions. The main concerns have been adequately resolved. The current version can be considered for acceptance.
We thank the reviewer for the critical analysis of our manuscript. We have reorganized and numbered the sections throughout the manuscript to increase the focus and readability.
Round 3
Reviewer 1 Report
Comments and Suggestions for Authors
The Authors have addressed my concerns.
The manuscript is now acceptable for publication in IJMS.